

# EsoDetect: computational validation and algorithm development of a novel diagnostic and prognostic tool for dysplasia in Barrett's esophagus

Migla Miskinyte, Benilde Pondeca, José B. Pereira-Leal and Joana Cardoso

Ophiomics, Lisbon, Portugal

## ABSTRACT

Barrett's esophagus (BE) is the only known precursor to esophageal adenocarcinoma (EAC), a malignancy with increasing incidence and unfavorable prognosis. This study endeavors to identify BE biomarkers capable of diagnosing low-grade dysplasia (LGD) in BE, as well as biomarkers that can predict the progression from BE to EAC to be subsequently integrated into diagnostic and prognostic algorithms. Datasets containing gene expression data from metaplastic and dysplastic BE, as well as EAC tissue samples, were collected from public databases and used to explore gene expression patterns that differentiate between non-dysplastic (ND) and LGD BE (for diagnostic purposes) and between non-progressed and progressed BE (for prognostic purposes). Specifically, for the diagnostic application, three RNAseq datasets were employed, while for the prognostic application, nine microarray datasets were identified, and 25 previously described genes were validated. A thresholding function was applied to each gene to determine the optimal gene expression threshold for group differentiation. All analyzed genes were ranked based on the F1-score metrics. Following the identification of genes with superior performance, different classifiers were trained. Subsequently, the best algorithms for diagnostic and prognostic applications were selected. In evaluating the value of gene expression for diagnosis and prognosis, the analyzed datasets allowed for the ranking of biomarkers, resulting in eighteen diagnostic genes and fifteen prognostic genes that were used for further algorithm development. Ultimately, a linear support vector machine algorithm incorporating ten genes was identified for diagnostic application, while a radial basis function support vector machine algorithm, also utilizing ten genes, was selected for prognostic prediction. Notably, both classifiers achieved recall and specificity scores exceeding 0.90. The identified algorithms, along with their associated biomarkers, hold significant potential to aid in the early management of malignant progression of BE. Their strengths lie in their development using multiple independent datasets and their ability to demonstrate recall and specificity levels superior to those reported in the existing literature. Ongoing experimental and clinical validation is essential to further substantiate their utility and effectiveness, and to ensure that these tools can be reliably integrated into clinical practice to improve patient outcomes.

Corresponding author
Joana Cardoso, jvaz@ophiomics.com

## INTRODUCTION

Barrett's esophagus (BE) is characterized by the replacement of the normal squamous epithelium lining the lower esophagus with specialized columnar cells (intestinal metaplasia) (*Choi & Sanagapalli, 2022*; *Killcoyne & Fitzgerald, 2021*; *Klavan et al., 2018*; *Spechler & Souza, 2014*). This transformation occurs because of chronic gastroesophageal reflux disease (GERD) (*Choi & Sanagapalli, 2022*; *Spechler & Souza, 2014*) and exposure to stomach acid (*Klavan et al., 2018*). Approximately 10% of patients with GERD are likely to progress to a diagnosis of BE over 5 years (*Malfertheiner et al., 2012*). Individuals with BE have a significantly increased risk of developing esophageal adenocarcinoma (EAC). Typically, the progression of EAC starts with GERD, followed by abnormal columnar cells characteristic of BE, which, over time, can progress to dysplasia and eventually become EAC. Despite BE's role as a precursor to EAC, the exact risk factors associated with BE are still not fully understood but include age ($\geq$60–70 years), male gender (*Fabian & Leung, 2021*), tobacco use (*Cook et al., 2012*; *Sinha, Abdulkader & Gupta, 2016*), obesity (*Kamat et al., 2009*), and hiatal hernia (*Andrici et al., 2013*).

The clinical relevance of BE relies on its role as the sole known precursor lesion for EAC (*Mittal et al., 2021*; *Spechler & Souza, 2014*). This specific type of esophageal cancer constitutes already around two-thirds of all cases of esophageal cancer in high-income countries (*Sung et al., 2021*), with 85,700 new EAC cases estimated worldwide in 2020. Over the next two decades, a staggering 65% increase (equivalent to approximately 55,600 additional cases annually) is predicted (*Morgan et al., 2022*). EAC is a major problem because of its association with poor survival rates, one of the lowest in oncology. Post-diagnosis, EAC presents a 23% 5-year survival and a median survival of only 15 months (*Then et al., 2020*), highlighting the need for efficient methods for EAC management. This low survival is mainly due to late diagnosis, limited treatment options, poor prognosis, high rate of early metastasis, and difficulties in early detection (*Fabian & Leung, 2021*).

Due to the low progression rate of BE to EAC (estimates 0.1–0.5, reviewed by *Hamade et al. (2019)*), most BE patients never progress to cancer. However, GERD is becoming increasingly prevalent, with a global estimate of 783 million prevalent cases in 2019 (*Li, Hoefnagel & Krishnadath, 2023*). Factors like population growth, aging, lifestyle changes, and improved living standards contribute to the rising incidence of GERD (*Zhang et al., 2022a*; *Zhang et al., 2022b*). As BE is a complication of GERD and a significant risk factor for EAC, the increasing prevalence of GERD cases represents a menace to future management of EAC. The increased prevalence of GERD leads to a higher incidence of BE cases and pressures for BE screening and diagnosis, resulting in a significant economic burden for patients, families, health services, and society.

Currently, BE serves as a critical warning sign and its surveillance is essential for effective risk stratification. BE screening and surveillance methods involve endoscopic sampling of biopsies from four quadrants according to the Seattle biopsy protocol (*Lee et al., 2018*; *Spechler et al., 2011*) followed by histological analysis to classify detectable BE lesions as non-dysplastic (NDBE), indefinite for dysplasia (IND), low-grade dysplasia (LGD), or high-grade dysplasia (HGD) (*Mittal et al., 2021*). Limitations to the success of current

strategies include but are not limited to, difficulties with endoscopic identification of dysplasia, biopsy sampling error, low inter-observer reproducibility in histologic assessment of dysplasia among pathologists, lack of reliable biomarkers, access to specialized care and patient compliance (*Eluri & Shaheen, 2017*). Variability in the endoscopic and histologic assessment are commonly known issues: BE endoscopic/pathological management is time-consuming and depends on the clinical experience of the physicians involved in the endoscopic examination and/or histological analysis—who are mostly available in BE reference centers. For example, one meta-analysis reported up to 25% and 24% of EACs were respectively missed during surveillance or when the analysis was restricted to NDBE patients (*Visrodia et al., 2016*). Regarding histological analysis, the inter-observer agreement among pathologists has been reported as only 58% when it comes to distinguishing normal esophagus from BE and was even lower (less than 50%) when diagnosing LGD in BE patients (*Runge, Abrams & Shaheen, 2015*; *Sharma, 2004*). The lack of agreement can become particularly problematic when many cases of BE are classified as IND (60% of dysplastic cases in a study by *Alshelleh et al., 2018*) and when the interobserver agreement is even poorer than for LGD (*Thota et al., 2016*).

There is emerging evidence that the addition of biomarkers to risk stratification models could increase BE diagnostic accuracy compared to current surveillance methods (*Shaheen et al., 2022*). These biomarkers range from the incorporation of more clinical variables (*Galipeau et al., 2007*; *Vaughan, Onstad & Dai, 2019*) to molecular features such as genomic instability (*Maley et al., 2004*; *Merlo et al., 2010*; *Mokrowiecka et al., 2012*; *Paulson et al., 2009*; *Trindade et al., 2019*), gene expression patterns (*Cardoso et al., 2016*; *Selaru et al., 2022*), epigenetics (*Jin et al., 2009*; *Moinova et al., 2018*), and proteomics (*Abdo et al., 2018*). In addition to biomarkers, the recent emergence of artificial intelligence (AI) tools opens the prospect of improving the effectiveness of BE diagnosis and surveillance. A recent meta-analysis revealed that deep learning algorithms applied to endoscopy images in the surveillance of BE-related neoplasia are highly accurate (pooled sensitivity and specificity of 90.3% and 84.4%, respectively) in detecting early HGD/EAC (*Tan et al., 2022*), despite the absence of data for LGD. However, most diagnostic and prognostic tools (biomarkers, AI), still lack substantial validation in large patient cohorts, refraining from their usage in clinical practice (*Fouad et al., 2014*). In addition, the new tools available do not reach yet maximum performance. For example, when predicting the neoplastic progression to HGD/EAC, both TP53 staining and Tissue Cypher test demonstrate high specificity (86% and 82%, respectively) but to the detriment of low sensitivity/recall (49% and 55%, respectively) (reviewed by *Honing & Fitzgerald (2023)*).

While it is not yet clear whether regular surveillance surely leads to earlier detection of dysplasia and consequently to a decrease in mortality from EAC (*Mejza & Małecka-Wojciesko, 2023*) surveillance is still the only recommended strategy for BE and EAC management. There is room for new diagnostic and prognostic tools to support clinicians when diagnosing BE dysplasia and segmenting patients based on the risk of BE progression to EAC.

The current study explores the diagnostic and prognostic value of gene expression patterns from BE tissue samples from public datasets in the context of BE. Envisioning its

clinical applicability, it aims to identify biomarkers that can accurately identify dysplasia within BE lesions (diagnostic application) and biomarkers that can predict the progression to EAC (prognostic application). It is also intended to understand the individual and combined predictive value of each selected biomarker in both contexts through their implementation using machine learning algorithms.

## MATERIALS AND METHODS

### Dataset search

An exhaustive search for public datasets containing gene expression data related to BE, including normal esophageal epithelium, NDBE, BE with different degrees of dysplasia (LGD and HGD) and EAC was performed in the following databases: PubMed (https://pubmed.ncbi.nlm.nih.gov/), Gene Expression Omnibus (GEO, https://www.ncbi.nlm.nih.gov/geo/), Sequence Read Archive (SRA, https://www.ncbi.nlm.nih.gov/sra), and European Genome-Phenome Archive (https://ega-archive.org/). For the diagnostic application, the aim was to distinguish between NDBE and LGD BE. For the prognostic application, non-progressed BE (nonP-BE) and progressed BE (P-BE) data was studied. P-BE was defined as a BE adjacent to EAC. A summary of the methodology used is represented in Fig. 1 and described in detail below.

### Data pre-processing

In this study, raw RNA-seq data from projects GSE193946, GSE58963, and E-MTAB-4054 were obtained from the Sequence Read Archive (SRA) and the European Genome-Phenome Archive (EGA). We processed the data using a Docker environment equipped with Kallisto version 0.46.1 (docker image: jlnetosci/kallisto:v0.46.1), which facilitated the pseudo-alignment of the reads against the Homo_sapiens.GRCh38.cdna.all.release-107 reference transcriptome from Ensembl. Post-alignment, the transcript abundance estimates generated by Kallisto were imported into the R programming environment using the tximport package. This allowed transcript-level data to be transformed into gene-level counts, which were subsequently analyzed for differential expression. The combined data was filtered for low-expressed genes using the filterByExpr function in EdgeR (*Robinson, McCarthy & Smyth, 2009*), resulting in a dataset of 20,608 genes for downstream analysis. Samples were then normalized using the Trimmed Mean of M-values (TMM) normalization method and differential expression analysis was performed using EdgeR. For downstream analysis, including feature selection and classifier training, log-transformed CPM normalized values were used, which were subsequently corrected for batch effects using the ComBat function from the sva package (*Leek et al., 2012*).

In this study, microarray data was sourced from the Gene Expression Omnibus (GEO) database using the GEOquery package available in the R software. The data included accessions GSE1420, GSE36223, GSE13083, GSE37200, GSE34619, GSE26886, GSE39491, GSE100843, and an additional dataset from *Watts et al. (2007)*. Data was loaded and normalized using both the affy and oligo packages in R, depending on the array platform. The CEL files were read and processed using the frma function for robust multi-array average (RMA) normalization. Probe-level data was annotated and collapsed to gene-level

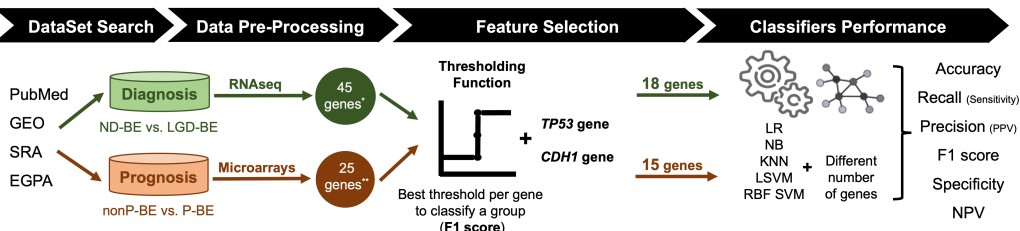

**Figure 1** **Methodology summary.** Datasets of interest were downloaded from public databases, such as PubMed, GEO (Gene Expression Omnibus), SRA (Sequence Read Archive), and EGPA (European Genome-Phenome). For the diagnostic application, *i.e.,* the distinction between non-dysplastic (ND) BE and low-grade dysplasia (LGD) BE, RNAseq datasets were used. Low-expression genes were excluded, resulting in a pre-selection of 45 genes. For the prognostic application, *i.e.,* the distinction between non-progressed Barrett's Esophagus (nonP-BE) and progressed-BE (P-BE), microarray datasets were identified, and 25 previously described genes were selected (*Killcoyne & Fitzgerald, 2021*) **. A Thresholding Function was applied to each gene to define the best gene expression threshold for group distinction. All analyzed genes were ranked by F1 score, and additional feature selection methods were applied for diagnostic genes*, determining the top genes for diagnosis and prognostic application. Due to their biological functions, two extra genes—*TP53* and *CDH1*—were added to both diagnostic and prognostic data sets, summing 18 diagnostic and 15 prognostic genes. Different algorithms—logistic regression (LR), naive Bayes (NB), K-nearest neighbours (KNN), linear support vector machines (LSVM), and radial basis function support vector machines (RBF SVM)—were trained using different numbers of genes.

data using Bioconductor annotation packages hugene10sttranscriptcluster.db, hgu133a.db, hgu133plus2.db, and hgu133a2.db along with the WGCNA package. Finally, the resulting gene expression data was merged into a single dataset for downstream analysis, with additional annotations indicating BE progression status. For prognostic application, 25 genes selected in previous work to distinguish nonP-BE from P-BE (*Cardoso et al., 2016*), were used in this study—*ACTN1, C1S, CCN1* (alias *CYR61*), *CDH1, CEBPB, CEBPD, COL4A1, CTSB, DKK3, DUSP1, IER3, JUN, LAMC1, PLPP3, RBPMS, SNAI1, SNAI2, SPARC, TNS1, TRMT112, TP53, TWIST1, VWF, WWTR1* (alias *TAZ*) and *ZEB1*. Box plots representing normalized expression values were generated using the ggplot2 (v3.4.0) and ggsignif (v0.6.4) R packages. Statistical analysis was performed using one-way ANOVA, followed by a *post hoc* Tukey's 'Honest Significant Difference' test, both from the R stats package (v4.1.1). When ANOVA assumptions were not met, a Kruskal–Wallis Rank Sum Test (R stats package v4.1.1) was performed, followed by a *post hoc* Dunn's Kruskal-Wallis Multiple Comparisons test (FSA R package v0.9.3). The significance threshold was set at $p$-value $< 0.05$.

## Threshold selection and determination of individual predictive power

For the distinction between NDBE and LGD (diagnostic) or nonP-BE and P-BE (prognostic) a Thresholding function was applied to the expression levels of each selected gene to determine an expression threshold. Performance metrics such as accuracy, recall (or sensitivity), precision (or positive predictive value—PPV), specificity, negative predictive value (NPV), and false positive rate (FPR) were calculated for each threshold, considering the known class of the samples. For the diagnostic application, other feature selection methods (Lasso, mutual information (MI) criteria, recursive feature elimination (RFE),

SelectKBest) were also applied to narrow down the most informative features that appeared at least twice in one of the methods. The threshold that yielded the highest F1-score was selected. Based on this metric, genes were ranked and the top 16 (diagnostic) and top 13 (prognostic) were considered for downstream analysis. Two additional genes—*TP53* and *CDH1*—were also included in the downstream analysis of both prognostic and diagnostic gene sets.

## Algorithmic analysis and evaluation of performance metrics

Gene expression values were used for algorithm training. Several classes of classifiers, with shown applicability to microarray and RNAseq data (*Jabeen, Ahmad & Raza, 2018*; *Peixoto et al., 2023*; *Pirooznia et al., 2008*), such as logistic regression (LR), naive Bayes (NB), K-nearest neighbours (KNN), and support vector machines (SVM) (with linear and radial basis function kernels), were implemented with default hyperparameters in Python programming language (v3.10.0), using the scikit-learn package (v1.0.1). We specifically chose LR (linear, interpretable), NB (probabilistic, fast), KNN (distance-based, non-parametric), SVM (linear margin-based), SVM (RBF kernel, nonlinear margin-based) models as they cover probabilistic, distance-based, linear and non-linear margin-based approaches which are widely used because of good performance on small, imbalanced gene-expression cohorts in the microarray/RNA-seq literature. A leave-one-out cross-validation procedure was used to evaluate the diagnostic or prognostic value of all possible combinations of genes (from $n = 2$ up to all selected diagnostic or prognostic genes). This involved leaving out one sample at a time for validation while using the remaining samples to create a balanced training set. The Synthetic Minority Oversampling Technique (SMOTE) was employed from the imbalanced-learn (v0.8.1) package. For LR, KNN, and SVM, features were standardized (scaled and centered) using scikit-learn's standard scaler module by subtracting the mean and scaling to the unit variance. Performance metrics such as accuracy, precision (PPV), recall (sensitivity), NPV, and precision and specificity were calculated and recorded for each full iteration of the validation strategy. The top-performing algorithms were chosen by maximizing performance metrics (accuracy, specificity, precision, recall, NPV, and F1-score, Table 1). The most frequent models, with the highest F1-score, were chosen to further select the best classifiers for both diagnostic and prognostic applications. The most frequently occurring genes (frequency ≥ 50 %) within the selected classifiers were chosen as features. Subsequently, the performance metrics were calculated using a decremental number of features, and the median value and standard deviation of each group of decremental subsets of genes were computed.

### *In-vitro* gene expression analysis
### *Cell culture*

Cell lines derived from metaplastic tissue (BAR-T and BAR-T10—from R. Souza, Baylor University Medical Center, Dallas, TX; *Jaiswal et al., 2007*; *Zhang et al., 2010*), dysplastic tissue (CP-B, CP-C and CP-D—from P. Rabinovitch, University of Washington, Seattle, WA; *Palanca-Wessels et al., 2003*), and EAC tissue (OE33, KYAE-1- from W. Dinjens, Erasmus Medical Center Cancer Institute, Rotterdam, Netherlands, and ESO26—*Boonstra et al., 2010*), were cultured in T75 flasks or 10-cm dishes (for metaplastic cells) until they

**Table 1   Best algorithm performance by metric maximization.**

| Application | Rank by | N. algorithms | Type of algorithm | Recall | Precision | F1 - score | Specificity | NPV | Accuracy |
|---|---|---|---|---|---|---|---|---|---|
| **Diagnostic** | Recall | 4,871 | KNN ($n = 196$) LSVM ($n = 2,426$) LR ($n = 124$ RBF SVM ($n = 2,125$) | 0.99 | 0.78–0.98 | 0.88–0.99 | 0.48–0.95 | 1.00 | 0.82–0.98 |
| | Precision | 3,050 | KNN ($n = 2,290$) LSVM ($n = 472$) LR ($n = 259$) NB ($n = 21$) RBF SVM ($n = 8$) | 0.65–1.00 | 0.97 | 0.79–0.99 | 0.95–1.00 | 0.60–1.00 | 0.77–0.98 |
| | F1-score | 1881 | KNN ($n = 288$) LSVM ($n = 1,115$) LR ($n = 444$) RBF SVM ($n = 34$) | 0.92–1.00 | 0.93–1.00 | 0.96 | 0.86–1.00 | 0.88–1.00 | 0.95–0.98 |
| | Specificity | 231 | KNN ($n = 223$) LSVM ($n = 8$) | 0.65–0.95 | 1.00 | 0.79–0.97 | 0.99 | 0.60–0.91 | 0.77–0.97 |
| | NPV | 4871 | KNN ($n = 196$) LSVM ($n = 2,426$) LR ($n = 124$) RBF SVM ($n = 2,125$) | 1.00 | 0.78–0.98 | 0.88–0.99 | 0.48–0.95 | 0.99 | 0.82–0.98 |
| | Accuracy | 212 | KNN ($n = 38$) LSVM ($n = 157$) LR ($n = 16$) RBF SVM ($n = 1$) | 0.95–1.00 | 0.95–1.00 | 0.97–0.99 | 0.90–1.00 | 0.91–1.00 | 0.96 |
| **Prognostic** | Recall | 13 | LR ($n = 7$), LSVM ($n = 5$), RBF SVM ($n = 1$) | 0.97 | 0.69–0.70 | 0.81 | 0.79–0.80 | 0.98 | 0.85–0.86 |
| | Precision | 582 | RBF SVM ($n = 449$) KNN ($n = 24$) LSVM ($n = 17$) NB ($n = 92$) LR ($n = 348$) | 0.88–0.95 | 0.99 | 0.93–0.98 | 1.00 | 0.94–0.98 | 0.96–0.98 |
| | F1-score | 12,971 | RBF SVM ($n = 5,794$) KNN ($n = 2,465$) LSVM ($n = 2,230$) NB ($n = 2,134$) LR ($n = 348$) | 0.92–0.95 | 0.97–1.00 | 0.96 | 0.99–1.00 | 0.96–0.98 | 0.98 |
| | Specificity | 8,430 | KNN ($n = 586$) LSVM ($n = 569$) LR ($n = 38$) NB ($n = 1,953$) RBF SVM ($n = 5,284$) | 0.83–0.95 | 0.98–1.00 | 0.9–0.98 | 0.99 | 0.92–0.98 | 0.94–0.98 |
| | NPV | 13 | LR ($n = 7$), LSVM ($n = 5$), RBF SVM ($n = 1$) | 0.97 | 0.69–0.70 | 0.81 | 0.79–0.80 | 0.98 | 0.85–0.86 |
| | Accuracy | 2,404 | KNN ($n = 264$) LSVM ($n = 370$) LR ($n = 28$) NB ($n = 415$) RBF SVM ($n = 1,327$) | 0.94–0.95 | 0.98–1 | 0.97–0.98 | 0.99–1 | 0.98 | 0.98 |

**Notes.**

LR, logistic regression; LSVM, linear support vector machine; RBF SVM, radial basis function support vector machine; KNN, K-nearest neighbors; NB, Naïve Bayes.
Selected algorithms are highlighted in grey.

reached 80% to 90% confluence. They were then detached with 0.25% Trypsin-EDTA, neutralized with complete culture medium, and collected in 15 mL Falcon tubes by centrifugation at $300 \times$ g for 5 min at room temperature. The supernatant was discarded, and the pellet was washed twice with $1 \times$ PBS to remove residual medium and trypsin.

After washing, the pellets were transferred to two mL Eppendorf tubes and immediately frozen at −80 °C for long-term storage.

### RNA extraction

Cell pellets from the cell lines metaplasia, dysplasia and EAC were used to extract RNA using the RNeasy Mini Kit (#74104, Qiagen, Hilden, Germany), following the manufacturer's instructions.

For formalin-fixed paraffin-embedded (FFPE) tissue samples, RNA was isolated from two consecutive sections per sample, each approximately 20 mm$^2$ and five μm. Tissue samples were deparaffinized using the deparaffinization solution (#19093, Qiagen, Hilden, Germany) prior to RNA extraction with the RNeasy FFPE Kit (#73504, Qiagen, Hilden, Germany), according to the manufacturer's instructions (with one modification: proteinase K incubation was performed overnight). Samples and data from patients included in this study were provided by the Biobanks: Valdecilla (PT20/00067) and by the Biobank of the Aragon Health System (National Registry of Biobanks B. B.0000873) (PT20/00112), integrated in the Platform ISCIII Biobanks and Biomodels and they were processed following standard operating procedures with the appropriate approval of the Ethics and Scientific Committees.

All procedures involving human tissue samples were approved by the National Ethics Committee for Clinical Research—Comissão de Ética para a Investigação Clínica (CEIC), under approval number 2022_EO_24.

### Reverse transcription—quantitative real-time polymerase chain reaction (RT-qPCR)

For 1-Step RT-qPCR, reactions were performed in triplicate, using the TaqPath 1-step RT-qPCR Master Mix (#A15300, Thermo Fisher Scientific, Waltham, MA, USA) with a final reaction volume of 10 μL. Each reaction containing one μL of template, 0.25 μM of probe and 0.5 μM of each primer (Primers and probes used for RT-qPCR are listed in Table S2). Data acquisition and analysis were conducted using the QuantStudio Design & Analysis Software v1.5.1 software, using the cycling program: UNG incubation at 25 °C—2 min, Reverse Transcription at 50 °C—15 min, followed by Polymerase activation at 95 °C—2 min and 40 cycles of Amplification at 95 °C—3 s and 58 °C—30 s. To normalize gene expression levels, the geometric mean of the reference genes (*PGK1*, *ELF1*, and *RPL13A*) was subtracted from cycle threshold (Cq) of the target genes.

## RESULTS

### Diagnosis and prognosis dataset selection

For the development of the diagnostic application, 13 RNAseq-based datasets were identified, of which only three had publicly available clinical data—GSE58963 (*MacCarthy et al., 2014*), E-MTAB-4054 (*Maag et al., 2017*), GSE193946 (*Zhang et al., 2022a*; *Zhang et al., 2022b*)—and were therefore included in the present study. The BE data contained in each dataset is represented in Table 2. In total, data from 61 samples—comprising 21 NDBE, 40 LGD BE and 27 HGD—were included in the study.
**Table 2  Characterization of datasets for the diagnostic and prognostic applications.**

| Dataset | Diagnostic (RNAseq) | | | Prognostic (Microarray) | |
|---|---|---|---|---|---|
| | NDBE | LGD | HGD | nonP-BE | P-BE[*] |
| GSE1420 (*Moinova et al., 2018*) | – | – | – | 0 | 16 |
| *Watts et al. (2007)* and *Maag et al. (2017)* | – | – | – | 18 | 0 |
| GSE36223 (*Mokrowiecka et al., 2012*) | – | – | – | 23 | 0 |
| GSE13083 (*Morgan et al., 2022*) | – | – | – | 7 | 0 |
| GSE37200 (*Nancarrow et al., 2011*) | – | – | – | 0 | 46 |
| GSE34619 (*Odze, 2007*) | – | – | – | 10 | 0 |
| GSE26886 (*Ostrowski et al., 2007*) | – | – | – | 20 | 0 |
| GSE39491 (*Palanca-Wessels et al., 2003*) | – | – | – | 40 | 0 |
| GSE100843 (*Panda et al., 2021*) | – | – | – | 17 | 3 |
| GSE58963 (*Mejza & Małecka-Wojciesko, 2023*) | 7 | 7 | 7 | – | – |
| E_MTAB_4054 (*Merlo et al., 2010*) | 14 | 8 | – | – | – |
| GSE193946 (*Mittal et al., 2021*) | 0 | 25 | 20 | – | – |
| TOTAL N. samples | 21 | 40 | 27 | 135 | 65 |

**Notes.**

nonP-BE, non progressed Barrett's esophagus; P-BE, progressed Barrett's esophagus; NDBE, non-dysplastic Barrett's esophagus; LGD, low-grade dysplasia; HGD, high-grade dysplasia.

*P-BE was defined when a BE was adjacent to EAC.

For the prognostic application, 16 microarray datasets were identified, but only those generated on an Affymetrix platform were included in the downstream analysis to facilitate data merging. A total of nine microarray datasets were analyzed, including three previously analyzed by *Cardoso et al. (2016)*—GSE1420 (*Kimchi et al., 2005*), (*Watts et al., 2007*), and GSE13083 (*Stairs et al., 2008*) and six new ones, namely GSE36223 (*Ostrowski et al., 2007*), GSE37200 (*Silvers et al., 2010*), GSE34619 (*Di Pietro et al., 2012*), GSE26886 (*Wang, Ma & Kemmner, 2013*), GSE39491 (*Hyland et al., 2014*), and GSE100843 (*Cummings et al., 2017*). In total, data from 200 samples—representing 135 nonP-BE and 65 P-BE—were included in the study as shown in Table 2.

## Identification of differentially expressed genes in a diagnostic and prognostic setting

In this study, we aimed to identify diagnostic biomarkers that can distinguish between ND-BE and LGD-BE. For this purpose, we utilized three RNAseq datasets (as listed in Table 2). Low-expression genes were excluded from each dataset, resulting in the inclusion of 20,608 genes in our analysis. After normalization, we conducted differential expression analysis between LGDBE and NDBE (Fig. S1A), and HGDBE and NDBE (Fig. S1B) using EdgeR's quasi-likelihood approach. This approach accounted for disease staging and batch effects from the three different datasets as factors in the model (Table S3, Fig. S1C).

Following the differential expression analysis, we identified 30 biomarkers through a systematic selection process. First, we selected differentially expressed genes (DEGs) with an absolute log fold change (logFC) of ≥ 1 between LGDBE and NDBE, with a false discovery rate (FDR) of <0.05. From these DEGs, we filtered for genes that showed the same direction of expression change in the HGDBE *vs.* NDBE comparison (FDR < 0.05), resulting in 14 genes (Fig. S1A). Second, we identified DEGs in the HGDBE *vs.* NDBE comparison with an absolute logFC of ≥2 (FDR < 0.05). Among these genes, we selected those that also exhibited the same direction of expression change in the LGDBE *vs.* NDBE comparison (considering *p*-value < 0.05 for significance), resulting in 16 genes (Fig. S1B). This two-step filtering strategy ensured that the selected biomarkers not only had significant differential expression but also consistent expression patterns across different stages of disease progression. Given the strong batch effect observed (see Fig. S1), there was a risk of losing biologically relevant genes in the LGDBE *vs.* NDBE comparison due to this variation. To mitigate this problem, we also performed separate analyses of the EMTAB_4054 (Table S4) and GSE58963 (Table S5) datasets. We employed the glmRobust pipeline to independently identify differentially expressed genes between the LGDBE and NDBE groups within each dataset. From these separate analyses, we identified an additional 13 genes with an absolute logFC greater than 1 and an FDR < 0.05. These genes were consistently found in both datasets and exhibited the same direction of expression change (Fig. S2). Moreover, these genes showed consistent directional changes in the previous HGDBE *vs.* NDBE comparison. Thus, they were also included in the biomarker list (Table S6). Given their established role in the biology of BE and EAC, we also included two additional genes—*TP53* and *CDH1*—in the downstream analysis, resulting in a total of 45 candidate genes for distinguishing between NDBE and LGD.

For the prognostic set of biomarkers, we re-analyzed 25 genes that we had previously identified to have prognostic value (*Cardoso et al., 2016*), namely *ACTN1, C1S, CCN1* (alias *CYR61*), *CDH1, CEBPB, CEBPD, COL4A1, CTSB, DKK3, DUSP1, IER3, JUN, LAMC1, PLPP3, RBPMS, SNAI1, SNAI2, SPARC, TNS1, TP53, TRMT112, TWIST1, VWF, WWTR1* (alias *TAZ*) and *ZEB1*. For validation purposes, we added six independent datasets to the three datasets we originally analyzed. We observed significant differential gene expression (adj. *p-value* < 0.05) between P-BE and nonP-BE categories for most of the genes of interest, except for *CDH1, DKK3, SNAI2, and WWTR1*.

## Application of a thresholding function for the selecting genes with the highest predictive value

To each selected gene, we applied a Thresholding function, to determine a gene expression threshold for distinguishing different levels of gene expression between groups of samples with distinct diagnosis (NDBE *vs.* LGD-BE) or with distinct prognosis (nonP-BE *vs.* P-BE). We defined the best individual threshold of gene expression for each selected gene, 45 for diagnosis and 25 for prognosis, reflecting the individual predictive value of each gene. Genes were then ranked by the harmonic mean of recall and precision (F1-score) to ensure accurate selection. This procedure identified the top 15 genes for predicting the malignant progression of BE lesions with F1-score above 0.67 (Table S7). From the top 45 diagnostic

genes, including CDH1 and TP53, genes with higher expression values (log2CPM above 1) were filtered. To further refine a list of candidates, we used several feature selection methods: Lasso, MI criteria, RFE, and SelectKBest. Additionally, feature correlation analysis was conducted to identify and eliminate highly correlated features (Pearson's correlation coefficient > 0.9). Hence, for diagnostic purposes, we further narrowed down the selection to genes that were chosen at least twice in one of the feature selection methods and F1-score above 0.7, which identified the top 16 genes for diagnosing dysplasia in the context of BE (Table S8).

Building on our identification of the top diagnostic and top prognostic genes using the Thresholding function and various feature selection methods, the ROC curve results (Fig. 2) further validate their predictive power using a logistic regression classifier. In both diagnostic (Fig. 2A) and prognostic (Fig. 2B) contexts, most of the genes have area under the curve (AUC) values above 0.50 (random chance line), with some reaching individual values of 0.90, demonstrating a substantial predictive value of the selected genes in both diagnostic and prognostic applications.

## SVM algorithms were the best for both diagnostic and prognostic applications

The diagnostic and prognostic gene groups were utilized to train the most effective diagnostic and prognostic algorithms. Various classifiers—logistic regression (LR), naive Bayes (NB), K-nearest neighbors (KNN), linear support vector machine (LSVM), and radial basis function support vector machine (RBF SVM)—were examined using increasing combinations of genes, ranging from $n = 2$ up to the total number, for diagnostic and prognostic applications.

The algorithms were ranked based on their performance metrics for each application (see Table 1). However, no algorithms optimized all performance metrics for both applications. Nevertheless, the LSVM algorithms emerged as the best for diagnostic purposes, maximizing the F1-score and accuracy (refer to Fig. 3A and Table 1).

For the prognostic application, a similar trend was observed, where the RBF SVM type performed best according to the F1-score and accuracy metrics (refer to Fig. 3B and Table 1).

The study found that among the selected types of algorithms, those with an F1-score above 0.96 included 1,115 LSVM for diagnostic and 5,794 RBF SVM for prognostic. The analysis identified the most frequent genes (over 50 %) across the best-performing algorithm class (Tables S9 and S10). Ultimately, ten genes were selected for identifying LGD BE using a LSVM algorithm: *IGHV3-43, SLC38A4, PLLP, CELA3A, IGHV4-31, TMPRSS5, TP53, NR4A1, ATF3, IFI27*. For identifying P-BE, ten genes were selected using an RBF SVM algorithm: *SNAI1, C1S, DUSP1, CEBPB, COL4A1, ZEB1, CEBPD, CCN1, LAMC1* and *TWIST1*.

The performance of each selected algorithm (LSVM for diagnosis and RBF SVM for prognosis) was evaluated using the most frequent genes (10 for diagnosis and 10 for prognosis) as features. To test different random states while avoiding algorithm bias, 100 runs were performed for each algorithm with the same features. Table 3 presents

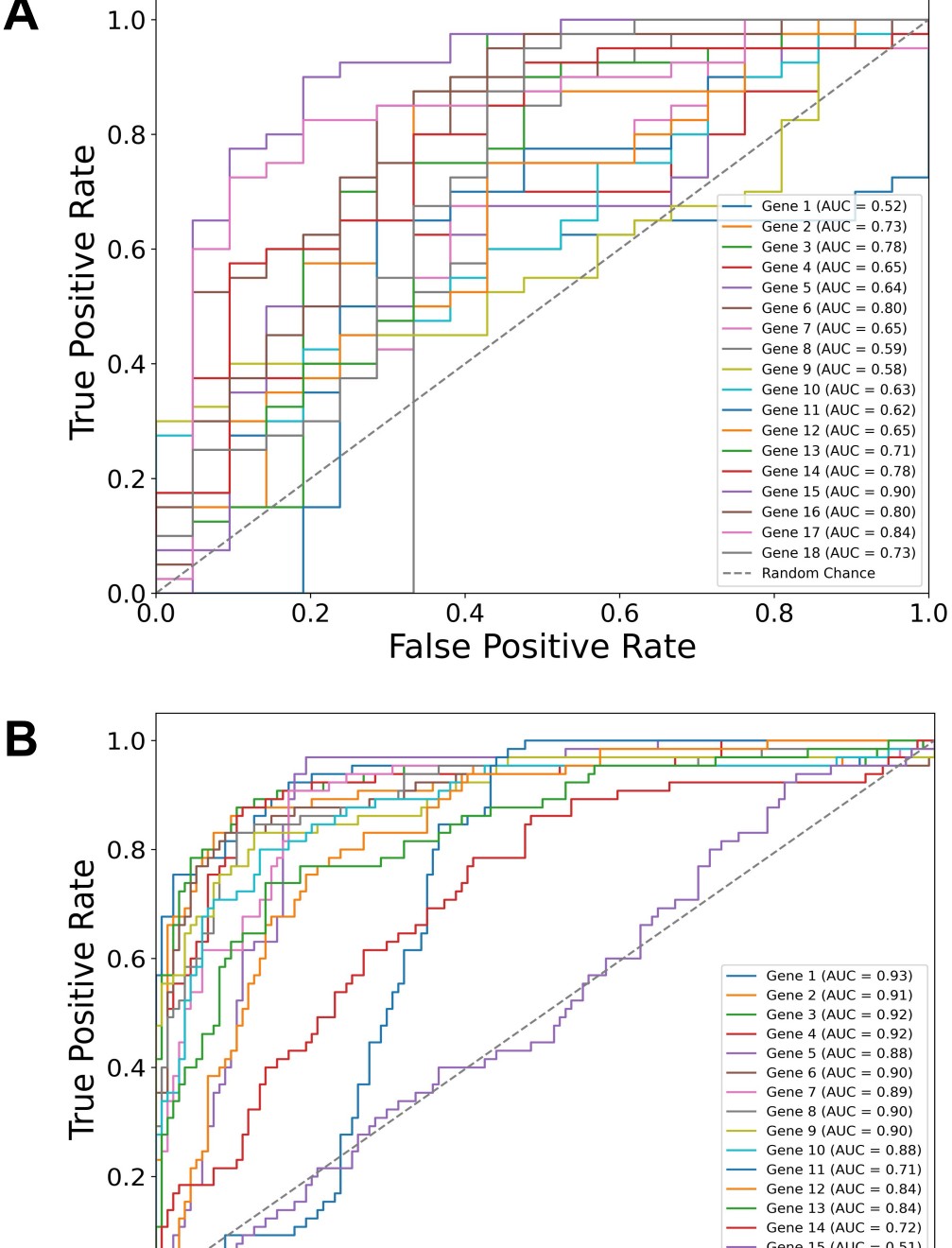

**Figure 2 Gene-specific ROC curves for diagnostic and prognostic predictions.** Receiver operating characteristic (ROC) curves for individual genes depicting their predictive value in two contexts: (A) Diagnostic (dysplasia) and (B) prognostic (progression) using a logistic regression classifier. The area under the curve (AUC) values for each gene are indicated in the legends. Notably, the predictive values of TP53 and CDH1 genes are also included, although they were manually added to the sets.

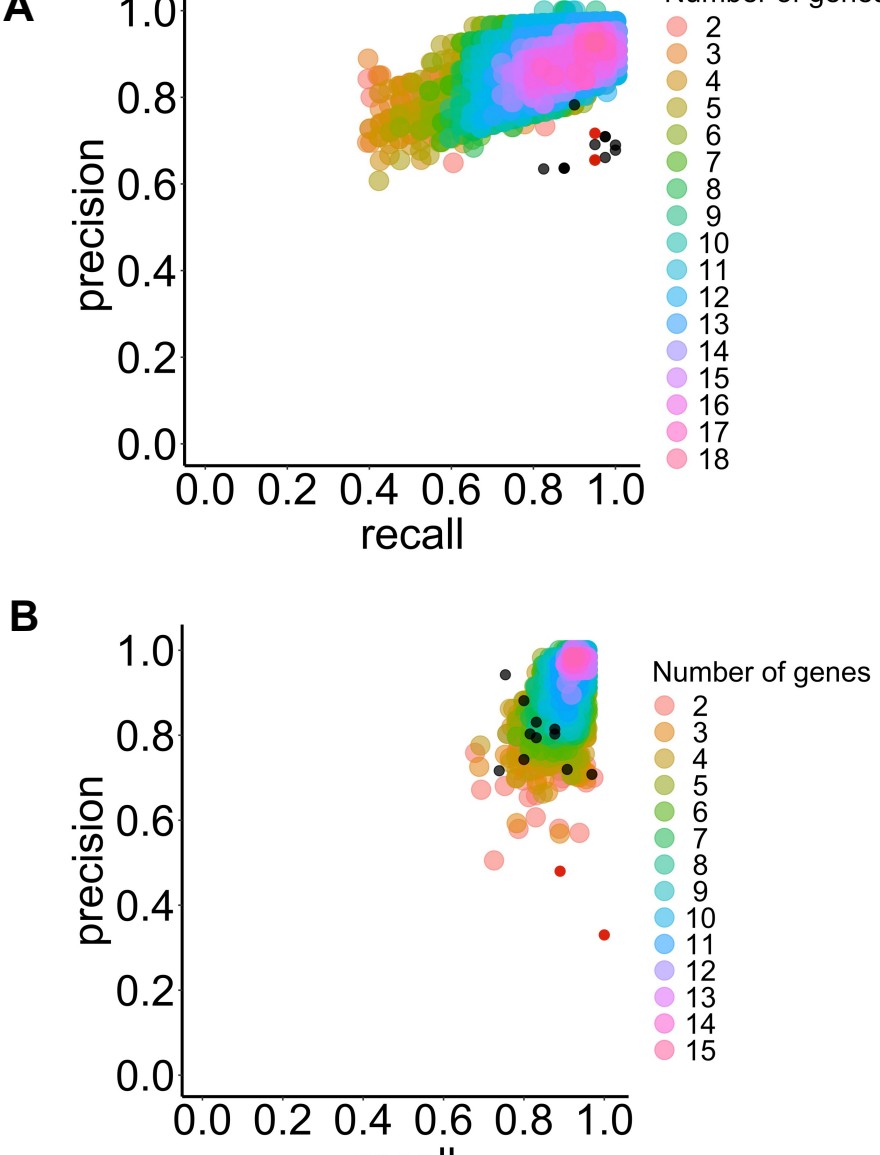

**Figure 3** **Precision and recall for the selected classifier type with increasing combinations of genes of interest to predict BE dysplasia (diagnostic) and BE malignant progression (prognostic).** This illustrates the performance of the chosen classifier types in the predicting dysplasia (A) and progression (B) when different numbers of genes of interest are combined (colored dots). The individual predictive value for the best threshold of each previously selected gene is also represented (black dots) for diagnostic (see Fig. 3A) and prognostic (see Fig. 3B). Colors represent different numbers of combined genes. LSVM (A), and RBFSVM (B). Red dots represent manually added CDH1 and TP53 genes.

**Table 3  Performance of the selected algorithms with the selected genes as features after 100 runs.**

| | A. Diagnostic | | | B. Prognostic | |
|---|---|---|---|---|---|
| | Mean | Standard deviation | | Mean | Standard deviation |
| Accuracy | 0,946 | 0,014 | Accuracy | 0,977 | 0,003 |
| Precision | 0,932 | 0,012 | Precision | 0,977 | 0,008 |
| Recall | 0,991 | 0,017 | Recall | 0,952 | 0,005 |
| F1 score | 0,960 | 0,010 | F1 score | 0,965 | 0,004 |
| TP | 39,630 | 0,677 | TP | 61,900 | 0,302 |
| FP | 2,900 | 0,541 | FP | 1,430 | 0,498 |
| TN | 18,100 | 0,541 | TN | 133,570 | 0,498 |
| FN | 0,370 | 0,677 | FN | 3,100 | 0,302 |
| NPV | 0,981 | 0,033 | NPV | 0,977 | 0,002 |
| Specificity | 0,862 | 0,026 | Specificity | 0,989 | 0,004 |
| FPR | 0,138 | 0,026 | FPR | 0,011 | 0,004 |
| MCC | 0,882 | 0,030 | MCC | 0,948 | 0,007 |
| Cohen's k | 0,878 | 0,030 | Cohen's k | 0,948 | 0,006 |

Notes.

TP, True Positive; FP, False Positive; TN, True Negative; FN, False Negative; NPV, Negative Predictive Value; FPR, False Positive Rate; MCC, Matthews correlation coefficient; Cohen's k, Cohen's kappa score.

the mean values and respective standard deviations (SD) for each performance metric. All performance metrics were above 0.90, except for specificity for the LSVM diagnostic algorithm. The low standard deviations (below 0.05) indicated an increase in the predictive value of each algorithm when the selected genes were combined.

Finally, the performance of the two algorithms was evaluated by gradually decreasing the number of selected genes (Fig. 4). The diagnostic algorithm showed a decrease in performance after the removal of just one gene (Fig. 4A). In contrast, the prognostic algorithm showed noticeable changes only after the removal of four genes (Fig. 4B).

### *In-vitro* validation of key diagnostic and prognostic biomarkers

We conducted a validation study of the panel of biomarkers to distinguish between different stages of BE progression. Specifically, we performed RT-qPCR analysis to compare the expression levels of these biomarkers in different cell lines: metaplasia (BAR-T and BAR-T10), dysplasia (CP-B, CP-C and CP-D), and EAC (OE33, KYAE-1 and ESO26). Each biomarker was tested with three technical replicates in each cell line.

For dysplasia diagnosis, we analyzed the expression of biomarkers in both metaplasia and dysplasia cell lines (Fig. 5). In evaluating EAC prognosis, we compared the expression levels between metaplasia and EAC cell lines (Fig. 6). Normalized expression values against reference genes (*PGK1*, *ELF1* and *RPL13A*) highlighted significant differences in key markers. For instance, biomarkers such as *IFI27* and *ATF3* differentiated metaplasia from dysplasia with statistically significant $p$-values ($p = 0.009$ and $p = 0.003$, respectively), revealing their potential utility in dysplasia diagnosis. Similarly, *CEBPB*, *SNAI1* and *CCN1* (alias *CYR61*) genes showed significant expression changes between metaplasia and EAC ($p$-values of 0,031, 0.022 and 0.038, respectively). This supports their relevance for EAC prediction. The observed differential expression patterns suggest that these biomarkers

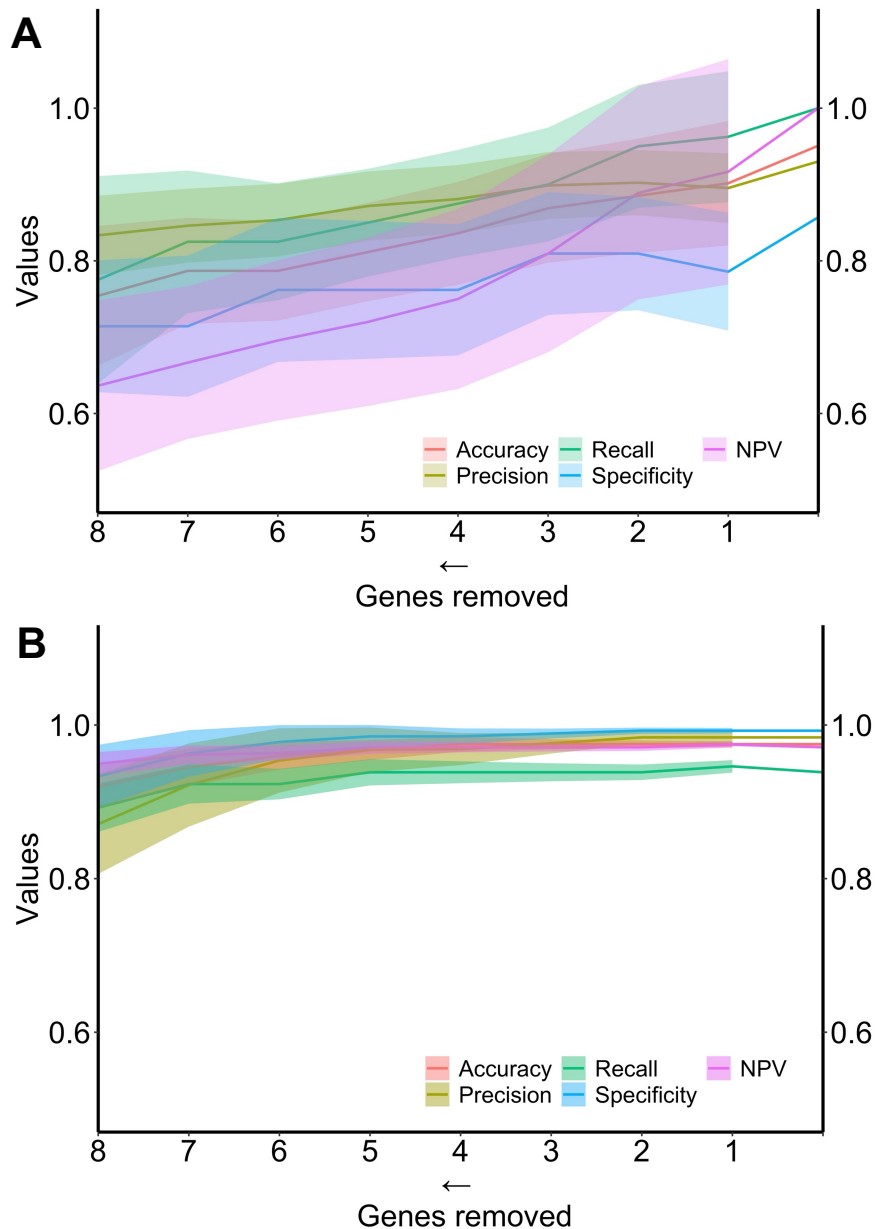

**Figure 4** **Metrics performance of the best algorithm in distinguishing ND-BE and LGD BE (diagnostic) and nonP-BE from P-BE (prognostic) when decreasing the number of genes included in the training.** Mean of each performance metric (solid lines) and its respective standard deviation (ribbons) for the diagnostic algorithm, linear support vector machine (LSVM) (A) and for the prognostic algorithm, radial basis function support vector machine (RBF SVM) algorithm (B) NPV—negative predictive value.

serve as valuable molecular tools for early detection of dysplasia and the risk of progression to EAC, facilitating timely clinical intervention.

Interestingly, some of the top-performing genes, namely *IGHV3-43*, *IGHV4-3* 1, *IGHV3-53*, and *PGC*, showed no detectable expression in cell lines. Since these genes ranked high

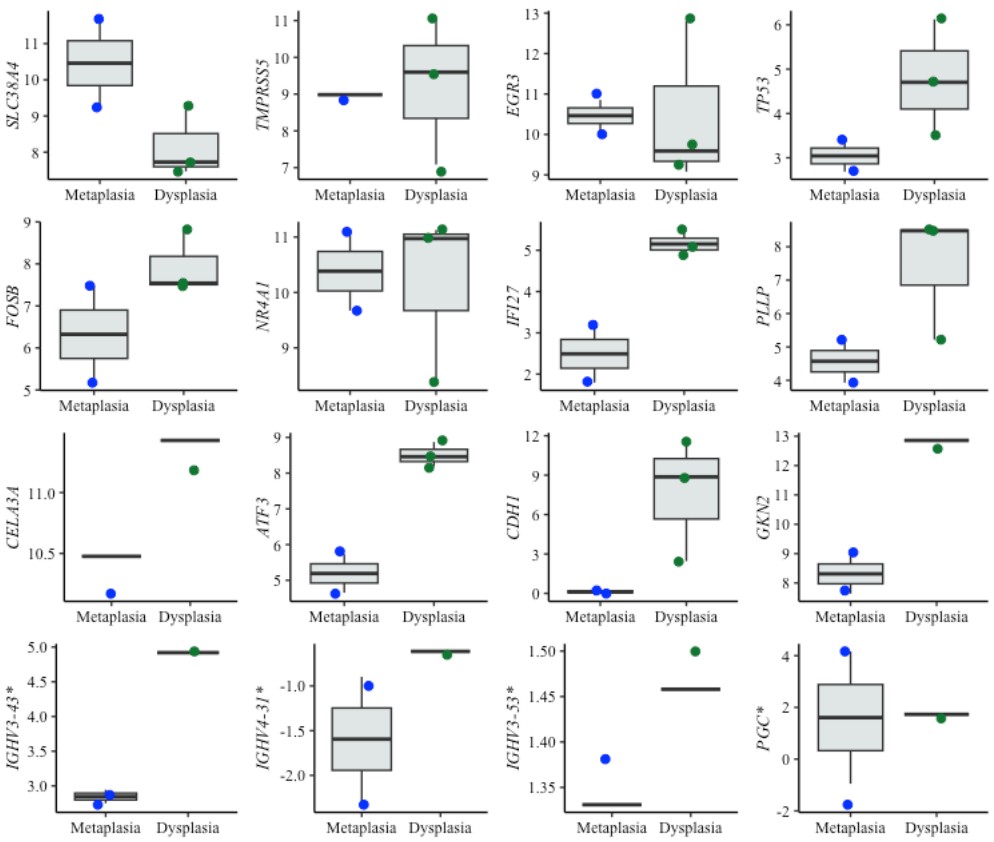

**Figure 5** **Boxplots of diagnostic genes expression levels.** Boxplots showing normalized expression levels of diagnostic genes across cell lines representing metaplasia (BAR-T and BAR-T10) and dysplasia (CP-B, CP-C, and CP-D). Gene expression levels were normalized to reference genes (PGK1, ELF1, and RPL13A). Blue and green dots represent individual expression values from metaplastic and dysplastic cell lines, respectively. *Genes were additionally validated in FFPE samples from patients diagnosed with BE, with and without dysplasia.

according to the diagnostic algorithm, we hypothesized that their expression originates from immune cells, typically absent in cell lines. To investigate this further, we specifically tested these genes in tissue samples from BE patients with and without dysplasia. Contrary to cell lines, the expression of these genes was detectable in patient samples, supporting the notion that immune cells- may play a critical role in BE progression (Fig. 5).

## DISCUSSION

BE is the only known precursor to EAC, a malignancy with rising incidence and poor prognosis. This underscores the need for more effective management methods, including assertive early diagnosis of dysplasia and prognostic prediction within BE surveillance programs. While tools incorporating biomarkers are continuously emerging, few have reached clinical validation and implementation. Even fewer combine biomarkers with AI and those under clinical validation or use, do not provide simultaneous detection of

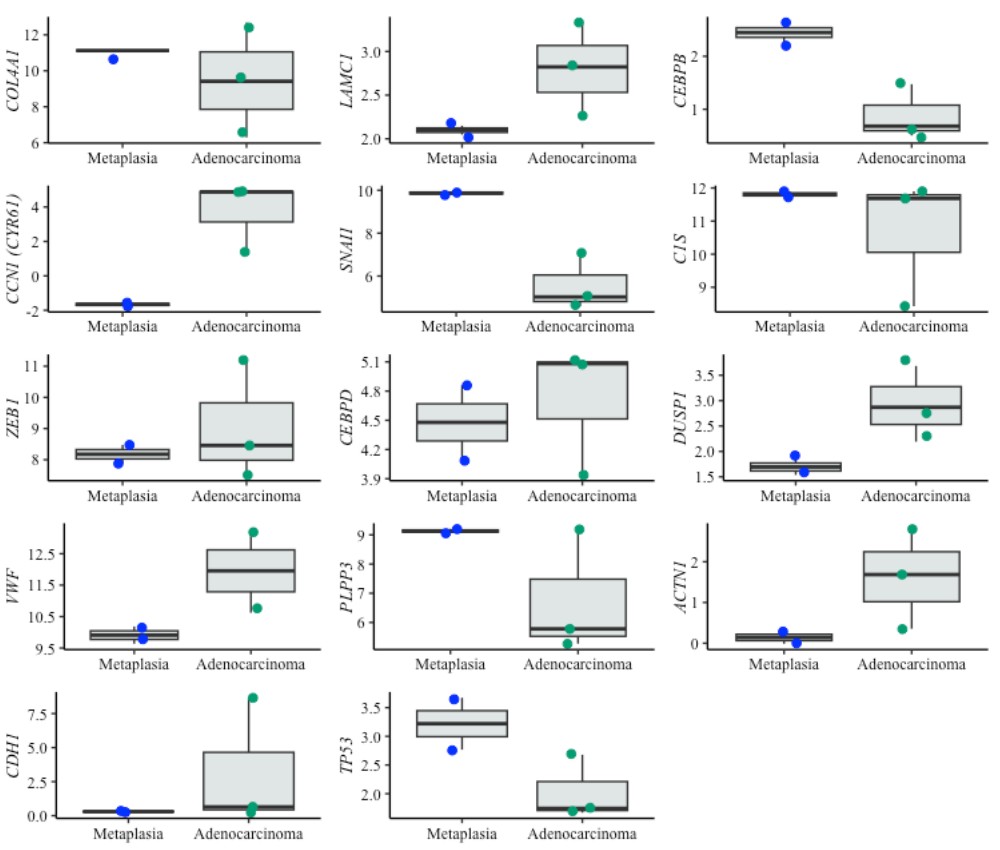

**Figure 6** **Boxplots of prognostic genes expression levels.** Boxplots showing normalized expression levels of prognostic genes across cell lines representing metaplasia (BAR-T and BAR-T10) and esophageal adenocarcinoma (OE33, KYAE-1, and ESO26). Gene expression levels were normalized to reference genes (PGK1, ELF1, and RPL13A). Blue and green dots indicate individual expression values from metaplastic and adenocarcinoma cell lines, respectively.

dysplasia and prognostic assessment. Moreover, none can simultaneously achieve high sensitivity (recall) and high specificity.

In this study, we developed two algorithms to assist with the diagnosis of dysplasia, the prognosis of BE, and ultimately the management of EAC. The genes of interest for dysplasia detection (diagnostic algorithm) were newly identified from the raw data of three different RNAseq datasets. Conversely, the algorithm developed for prognosis was based on a gene set identified in a previous study (*Cardoso et al., 2016*). For both applications, genes were ranked based on their F1 score, sensitivity (*aka* recall or true positive rate) and precision (*aka* positive predictive values) in predicting conditions such as LGD BE and P-BE.

In high-risk disease detection cases such as dysplasia, recall is a more important evaluation metric than precision because it can correctly identify all relevant positive cases (*i.e.,* samples containing dysplasia or at high risk of progressing to EAC). However, precision, which is the fraction of positive cases among all cases classified as positives by the model, is also crucial because it emphasizes the correctness of positive predictions made by the model (*i.e.,* measures how many cases are incorrectly classified as positive). In a

situation where false positives have significant implications, such as subjecting BE patients without dysplasia or with a low risk of progression to unnecessary treatments or screening intervention, precision matters. Since both high precision and high recall were desirable for the present study, the ranking was based on the F1 score, which combines precision and recall using their harmonic mean. Maximizing the F1 score implies maximizing both precision and recall simultaneously.

Performance metrics for each gene at its best threshold were high (see Tables S7 and S8). However, specificity and NPV were higher for the prognostic genes, showing their great potential to exclude patients who are not at risk for malignant progression.

To better explore the potential predictive value of the selected biomarkers, we trained machine learning algorithms testing all possible combinations of biomarkers in each gene set. The average metrics of the newly trained algorithms with combinations of biomarkers showed increased predictive power (Table 1) compared to the predictive power of individual genes (Tables S7 and S8), which is expected in the context of complex gene interactions. Finally, envisioning the clinical applicability of both algorithms, we evaluated the minimal number of biomarkers necessary to maintain high-performance metrics (LSVM for diagnosis and RBF SVM for prognosis) in each gene set. Both algorithms were tested with a decreasing number of genes, and as depicted in Fig. 4, a reduction in performance metrics was observed when removing one gene from the diagnostic set and four genes from the prognostic's gene set.

For diagnostic application, ten genes—*IGHV3-43*, *SLC38A4*, *PLLP*, *CELA3A*, *IGHV4-31*, *TMPRSS5*, *TP53*, *NR4A1*, *ATF3*, *IFI27*—were identified as the top candidates for dysplasia detection, particularly for distinguishing between NDBE and LGD BE. These genes are associated with different aspects of cancer biology, such as metabolism, cell invasion, and oncogenic processes, suggesting their potential as biomarkers in the context of BE dysplasia (*Jadhav & Zhang, 2017*; *Kastelein et al., 2013a*; *Li et al., 2021*; *Shulgin et al., 2021*). Moreover, transcription factors such as *NR4A1* and *ATF3*, have been previously associated with BE with LGD (*Maag et al., 2017*).

For the prognostic application, an RBF SVM algorithm was selected, which uses the expression pattern of ten genes (*SNAI1*, *C1S*, *DUSP1*, *CEBPB*, *COL4A1*, *ZEB1*, *CEBPD*, *CCN1*, *LAMC1* and *TWIST1*). Four of these genes—*SNAI1*, *COL4A1*, *ZEB1*, and *TWIST*—have been associated with epithelial-to-mesenchymal transition (*Lamouille, Xu & Derynck, 2014*). *COL4A1*, *ZEB1*, and *TWIST1* have also been described as potential screening biomarkers of BE malignant progression. *COL4A1* is upregulated in EAC *versus* BE (*Li et al., 2023a*; *Li et al., 2023b*; *Nancarrow et al., 2011*; *Qi et al., 2021*) and is associated with poor EAC prognosis (*Qi et al., 2021*), and it predicts the response to immune checkpoint inhibitors in EAC (*Li, Hoefnagel & Krishnadath, 2023*). Increased expression of *ZEB1* has been associated with the repression of *CDH1* (*Zhang et al., 2019*), which is associated with BE progression to EAC (*Darlavoix et al., 2009*; *Falkenback et al., 2008*; *Feith et al., 2004*; *Kalatskaya, 2016*; *Yao et al., 2021*). *TWIST1* up-regulation was observed in at-risk BE samples years before the emergence of any microscopic signs of malignancy (dysplasia/EAC) (*Cardoso et al., 2016*).

The genes *TP53* and *CDH1* were included in both gene sets to train the classifiers. *TP53* is known for its role in BE malignant progression (*Paulson et al., 2022*; *Pinto et al., 2022*), improved prediction of BE neoplastic progression (*Redston et al., 2022*), increased risk of dysplasia when abnormally expressed, and improved intra-observer agreement in dysplastic diagnosis (*Januszewicz et al., 2022*). *CDH1* has severely reduced or disorganized expression during BE dysplastic progression (reviewed by *Kalatskaya (2016)*) and an almost undetectable expression in poorly differentiated EAC (*Darlavoix et al., 2009*; *Falkenback et al., 2008*; *Feith et al., 2004*; *Kalatskaya, 2016*; *Yao et al., 2021*). While *TP53* alone is insufficient for diagnostic and for prognostic applications, it has been shown to have predictive value in combination with other biomarkers in the diagnostic setting. These findings confirm the previously studied role of *TP53* in the pathogenesis of BE dysplasia (*Kastelein et al., 2013b*; *Li, Hoefnagel & Krishnadath, 2023*). Because *TP53* mutations are often associated with a higher risk of progression in BE patients (*Redston et al., 2022*), further validation of this biomarker at the molecular level, including its mutational status and RNA expression levels, is warranted.

All metrics of both algorithms are higher when compared to currently available tools for risk stratification, such as *TP53* immunohistochemistry (0.49 recall/sensitivity, 0.86 specificity (*Kastelein et al., 2013b*) and TissueCypher (0.55 recall/sensitivity, 0.82 specificity for high-intermediate risk class 55%/82%) (*Eluri et al., 2015*; *Iyer et al., 2022*; *Jin et al., 2009*; *Moinova et al., 2018*). Tools for dysplasia detection, such as Wats3D and Cytosponge-*TFF3* are still under prospective evaluation. Wats3D provides an incremental yield of 7% for any dysplasia subtype but is negative for dysplasia in 62.5% of cases where an endoscopic biopsy confirmation to compare with the gold standard revealed dysplasia (*Codipilly et al., 2022*). The Cytosponge-*TFF3* test when combined with a multidimensional biomarker panel and fitted into a regression model was shown to be able to predict patients with dysplasia with good accuracy but further validation is still needed (*Ross-Innes et al., 2017*). Interestingly, in our top 45 genes for diagnostic application (Table S6), we have identified another trefoil factor, the *TFF2*, which is BE related gene.

A preliminary *in-vitro* validation of the selected diagnostic and prognostic biomarkers was conducted by examining their expression in metaplasia, dysplasia and EAC-derived cell lines. This validation confirmed their differential expression, highlighting their potential in distinguishing BE progression stages. Exceptionally, *IGHV3-43*, *IGHV4-31*, *IGHV3-53*, and *PGC* top-ranked genes were validated in FFPE samples from patients diagnosed with BE with and without dysplasia due to their lack of expression in the cell lines. The absence of immune cells in cell line cultures, which focus on epithelial cells, likely contributes to these findings. While we cannot exclude that the used cell lines may exhibit genetic differences from the original tissue, which potentially influences their molecular profiles (*Panda et al., 2021*), further clinical validation with a selected cohort of patient samples is warranted and is currently underway.

While imaging methods, such as hyperspectral imaging (HIS) has advanced real-time EAC detection, its utility to detect early LGD remains limited (*Wang et al., 2024*), molecular biomarkers could augment such modalities to create cost-effective, multimodal tools.

No molecular tools are currently implemented in clinical practice for identifying LGD/HGD BE. Dysplasia is a major biomarker in BE risk stratification, but it is often focal, making accurate characterization of collected BE biopsy challenging (*Odze, 2007*), and leading to many cases of BE classified as INDBE. INDBE is a management limbo for dysplasia, posing problems for clinicians. Most clinical tools developed for BE focus on risk stratification (prognosis) (*Abdo et al., 2018*; *Eluri et al., 2015*; *Iyer et al., 2022*; *Jin et al., 2009*; *Kaul et al., 2020*; *Moinova et al., 2018*; *Vaughan, Onstad & Dai, 2019*) and have a high specificity (identify and correctly exclude BE patients not at risk of progression). Simultaneously, these tools have a lower recall/sensitivity indicating their performance drops in detecting BE patients at true risk of progression.

New tests that aim for high recall and sensitivity are vital to avoid missing unacceptable true positive cases of LGD or HGD, as well as patients at risk of progression. However, these tests must also maintain high precision and high sensitivity to avoid incorrectly including patients not having dysplasia or having a low risk of progression. This balance can improve surveillance of high-risk patients while reducing unnecessary procedures for low-risk patients, ultimately lowering patient management costs.

Our approach, which combines machine learning algorithms with gene expression signatures, represents a promising breakthrough in healthcare. It has the potential to significantly enhance both the diagnosis and prognosis of dysplasia by delivering high recall and precision into clinical practice. Importantly, the required samples can be collected during routine endoscopy, an established procedure for patients diagnosed with BE, thereby minimizing any additional burden on patients or healthcare providers. Furthermore, the data processing and interpretation are fully automated by our algorithm, generating a clear and concise report for clinicians. This streamlined integration supports clinical decision-making while facilitating the broader adoption of molecular diagnostics in everyday medical practice.

## CONCLUSIONS

This study not only identified biomarkers and developed algorithms to detect LGD in BE biopsies and predict the progression of BE to EAC but also paved the way for creating new *in-vitro* laboratory tests for the diagnosis and prognosis of BE. Both algorithms were developed using datasets from public databases analyzing tissue samples obtained during routine endoscopy.

For the prediction of BE malignant progression, an LSVM algorithm featuring the identification of LGD was trained while an RBF SVM algorithm was trained for the prediction of BE malignant progression. Both algorithms reached high-performance metrics. To our knowledge, no existing tools can simultaneously detect dysplasia and assess the risk of progression with such high precision and recall.

Validation of the biomarkers and algorithms presented in this study in an independent test and validation patient cohort is currently under consideration. Additionally, while no other known risk factors (epidemiologic, clinical, histologic) have been combined with the presented biomarkers, incorporating patient demographic and clinical information

could further enhance the predictive value of the gene expression algorithms. Future algorithm developments will address this issue, demonstrating how such combinations can significantly boost their predictive power.

## ACKNOWLEDGEMENTS

We want to particularly acknowledge the patients and the Biobanks of Valdecilla (PT20/00067) and Aragon Health System (PT20/00112), integrated in the Platform ISCIII Biobanks and Biomodels for their collaboration.

### Funding

This work was supported by Ophiomics—Precision Medicine. The funders had no role in study design, data collection and analysis, decision to publish, or preparation of the manuscript.

### Grant Disclosures

The following grant information was disclosed by the authors:
Ophiomics—Precision Medicine.

### Competing Interests

The work described here is subject to European Patent Application No. 24172031.7; José B. Pereira-Leal and Joana Cardoso declare an ownership interest in the company Ophiomics—Precision Medicine. Migla Miskinyte and Benilde Pondeca are employees at Ophiomics—Precision Medicine.

### Author Contributions

- Migla Miskinyte conceived and designed the experiments, performed the experiments, analyzed the data, prepared figures and/or tables, authored or reviewed drafts of the article, and approved the final draft.
- Benilde Pondeca conceived and designed the experiments, performed the experiments, analyzed the data, prepared figures and/or tables, authored or reviewed drafts of the article, and approved the final draft.
- José B Pereira-Leal conceived and designed the experiments, authored or reviewed drafts of the article, and approved the final draft.
- Joana Cardoso conceived and designed the experiments, analyzed the data, authored or reviewed drafts of the article, and approved the final draft.

### Human Ethics

The following information was supplied relating to ethical approvals (i.e., approving body and any reference numbers):

All procedures involving human tissue samples were approved by the National Ethics Committee for Clinical Research –Comissão de Ética para a Investigação Clínica (CEIC), under approval number 2022_EO_24.

## Patent Disclosures

The following patent dependencies were disclosed by the authors:

The work described here is subject to European Patent Application No. 24172031.7.

## Data Availability

The RNA-seq data is available at Sequence Read Archive (SRA) and the European Genome-Phenome Archive (EGA): GSE193946, GSE58963, and E-MTAB-4054.

The microarray data is available at Gene Expression Omnibus (GEO) database: GSE1420, GSE36223, GSE13083, GSE37200, GSE34619, GSE26886, GSE39491, GSE100843.

The code is available at Zenodo: migaalee, & BenildePondeca. (2025). Ophiomics/Esodetect_publication: EsoDetect publication code (Esodetect_publication). Zenodo. https://doi.org/10.5281/zenodo.15302521.

## Supplemental Information

Supplemental information for this article can be found online at http://dx.doi.org/10.7717/peerj.19613#supplemental-information.

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
