# Peer review of "EsoDetect: computational validation and algorithm development of a novel diagnostic and prognostic tool for dysplasia in Barrett’s esophagus"

_PeerJ, doi:10.7717/peerj.19613_

## Round 0.1 · original submission · Major Revisions

I apologize that the review process took longer than expected. The manuscript has been examined by four referees, three of whom have raised major concerns. I hope you are able to address all of the reviewers' points satisfactorily with revisions to the manuscript and/or rebuttals in the response-to-review document.

·

Basic reporting

no comment.

Experimental design

The author should list the primers for the RT-qPCR.

Validity of the findings

no comment

Additional comments

Dear Author,
I sincerely appreciate the opportunity to review this manuscript. In the present study, the author focus on a novel diagnostic and prognostic tool for predict the progression from Barrett's esophagus (BE) to esophageal adenocarcinoma (EAC) by diagnostic and prognostic algorithms, and 25 biomarkers were validated. I think this research is well written, and this study may provide a powerful algorithms to aid in the early management of malignant progression of BE.

Reviewer 2 ·

Basic reporting

1. The authors call "in vivo" validation for cell line-based experiments. It should be called "in vitro."
2. All results of cell line experiments were included in supplementary materials. They should be included in main figures. As they investigated several cell lines, the expression levels should be demonstrated separately.
3. Cell culture conditions should be written in the method section.
4. The authors should consider publishing the algorithms used in this paper on GitHub following the Journal policy.

Experimental design

1. It is unclear why knowing the change in gene expression profiles through the esophageal metaplasia-dysplasia-adenocarcinoma sequence can overcome the current limitations, such as difficulties with endoscopic identification of dysplasia and biopsy sampling error as mentioned in the introduction section. Do they consider applying the results for less-invasive modalities like Cytosponge? The authors should more clearly state the clinical relevance of this investigation.
2. What kind of prognosis did the authors analyze? Overall survival? Please elaborate. In addition, the authors should compare the predictive performance of the prognosis between histological findings (low/high grade dysplasia) and gene expression-based algorithms. Use of Kaplan-Meier plot is encouraged to show the results.

Validity of the findings

no comment

Reviewer 3 ·

Basic reporting

• The authors detected diagnostic and prognostic biomarkers yet they need to describe their methods for avoiding biomarker discovery limitations of overfitting and selection bias. Additional functional studies should validate whether identified genes show biological significance even though identification was successful.

• To increase the motivation and need of the study with recent research on the An investigation into current research on Hyperspectral Imaging (HSI) development and its applications to esophageal cancer detection would strengthen the article.

1.Precision Imaging for Early Detection of Esophageal Cancer[https://www.mdpi.com/2306-5354/12/1/90]
2.Computer-aided endoscopic diagnostic system modified with hyperspectral imaging for the classification of esophageal neoplasms[https://pmc.ncbi.nlm.nih.gov/articles/PMC11646837/]
3.Evaluation of Spectrum-Aided Visual Enhancer (SAVE) in Esophageal Cancer Detection Using YOLO Frameworks[https://www.mdpi.com/2075-4418/14/11/1129]

Experimental design

• The research relies on public databases to perform its gene expression data analysis. The authors explain how their research handles the constraints from working with public databases which include inconsistent sample quality and various data collection protocols and possible data biases. The study can gain additional strength by including prospective data combined with additional independent datasets.

• The diagnostic and prognostic algorithms show reliable recall and specificity scores in their performance indicators. The authors provide strategies to confirm external clinical cohort uses for algorithm generalization and practical deployment ability.

• The study authors failed to explore how Hyperspectral Imaging (HSI) and other enhanced imaging solutions might improve Barrett's esophagus (BE) and esophageal adenocarcinoma (EAC) diagnosis along with prognosis. The authors should elaborate on their reasons for not selecting HSI or comparable imaging methods in combination with gene expression analysis for achieving improved diagnostic and prognostic capabilities.

Validity of the findings

• The algorithms demonstrate good diagnostic precision but the researchers need to examine what barriers exist when implementing these systems in medical care practice. What are the real-world obstacles to acquire gene expression data and implement this data into clinical operating procedures?

• The analysis uses support vector machine (SVM) algorithms for performing classifications. The authors have considered what performance metrics their algorithms demonstrate versus superior machine learning algorithms consisting of ensemble models or neural networks. The study would benefit from extra comparison results to affirm its findings.

Reviewer 4 ·

Basic reporting

All comments have been added in detail to the last section.

Experimental design

All comments have been added in detail to the last section.

Validity of the findings

All comments have been added in detail to the last section.

Additional comments

Review Report for PeerJ
(EsoDetect: Computational Validation and Algorithm Development of a Novel Diagnostic and Prognostic Tool for Dysplasia in Barrett’s Esophagus)

1. This study aims to identify biomarkers capable of diagnosing low-grade dysplasia in Barrett’s esophagus and predicting its progression to esophageal adenocarcinoma, developing machine learning-based diagnostic and prognostic models within the scope of the study.

2. In the introduction, Barrett’s esophagus, the importance of the subject and a certain level of literature are mentioned. However, in this section, a literature table should be added so that the literature related to the study can be more understandable, and then the main contributions of the study, differences from this literature, etc. should be added to the end of the introduction in more explanatory and itemized form.

3. In the study, open source datasets were determined by using four different databases. When the dataset and preprocessing steps preferred in the study are examined in detail, it is observed that they are at a sufficient level for the study.

4. It is noteworthy that five different machine learning models are preferred for classification. Although there are many different machine learning models in the literature that can be used to solve such problems, why these are preferred should be stated more clearly.

5. Although the types of evaluation metrics are basically suitable, it is recommended to analyze the results in more detail in terms of Cohen's Cappa and Matthews correlation coefficient (MCC) scores.

In conclusion, the study is important in terms of subject, but the sections listed above should be addressed completely.

---

## Round 0.2 · accepted · Accept

The manuscript was examined by four referees in the first round of review. In their revised version, the authors have addressed the referees' concerns to my satisfaction.